# Clinical Presentation of Parvovirus B19 Infection in Adults Living with HIV/AIDS: A Case Series

**DOI:** 10.3390/v15051124

**Published:** 2023-05-08

**Authors:** Daniela P. Mendes-de-Almeida, Joanna Paes Barreto Bokel, Arthur Daniel Rocha Alves, Alexandre G. Vizzoni, Isabel Cristina Ferreira Tavares, Mayara Secco Torres Silva, Juliana dos Santos Barbosa Netto, Beatriz Gilda Jegerhorn Grinsztejn, Luciane Almeida Amado Leon

**Affiliations:** 1Hematology Department, Evandro Chagas National Institute of Infectious Diseases, Oswaldo Cruz Foundation (FIOCRUZ), Rio de Janeiro 21040-360, RJ, Brazil; daniela.almeida@ini.fiocruz.br (D.P.M.-d.-A.); joanna.bokel@ini.fiocruz.br (J.P.B.B.); alexandre.vizzoni@ini.fiocruz.br (A.G.V.); 2Research Center, Instituto Nacional de Câncer (INCA), Rio de Janeiro 20220-430, RJ, Brazil; 3Department of Medical Affairs, Clinical Studies, and Post-Registration Surveillance (DEAME), Institute of Technology in Immunobiologicals/Bio-Manguinhos, Oswaldo Cruz Foundation (FIOCRUZ), Rio de Janeiro 21040-360, RJ, Brazil; 4Onco-Hematology Unit, Clínica São Vicente, Rio de Janeiro 22451-100, RJ, Brazil; 5Laboratory of Technological Development in Virology, Instituto Oswaldo Cruz, Oswaldo Cruz Foundation (FIOCRUZ), Rio de Janeiro 21040-360, RJ, Brazil; arthur.alves@ioc.fiocruz.br; 6Laboratory of Clinical Research on STD/AIDS, Evandro Chagas National Institute of Infectious Diseases, Oswaldo Cruz Foundation (FIOCRUZ), Rio de Janeiro 21040-360, RJ, Brazil; isabel.tavares@ini.fiocruz.br (I.C.F.T.); mayara.secco@ini.fiocruz.br (M.S.T.S.); juliananetto@gmail.com (J.d.S.B.N.); beatriz.grinsztejn@gmail.com (B.G.J.G.)

**Keywords:** Parvovirus B19, HIV infection, hemolysis, hereditary spherocytosis

## Abstract

Parvovirus B19 (B19V) infection varies clinically depending on the host’s immune status. Due to red blood cell precursors tropism, B19V can cause chronic anemia and transient aplastic crisis in patients with immunosuppression or chronic hemolysis. We report three rare cases of Brazilian adults living with human immunodeficiency virus (HIV) with B19V infection. All cases presented severe anemia and required red blood cell transfusions. The first patient had low CD4^+^ counts and was treated with intravenous immunoglobulin (IVIG). As he remained poorly adherent to antiretroviral therapy (ART), B19V detection persisted. The second patient had sudden pancytopenia despite being on ART with an undetectable HIV viral load. He had historically low CD4^+^ counts, fully responded to IVIG, and had undiagnosed hereditary spherocytosis. The third individual was recently diagnosed with HIV and tuberculosis (TB). One month after ART initiation, he was hospitalized with anemia aggravation and cholestatic hepatitis. An analysis of his serum revealed B19V DNA and anti-B19V IgG, corroborating bone marrow findings and a persistent B19V infection. The symptoms resolved and B19V became undetectable. In all cases, real time PCR was essential for diagnosing B19V. Our findings showed that adherence to ART was crucial to B19V clearance in HIV-patients and highlighted the importance of the early recognition of B19V disease in unexplained cytopenias.

## 1. Introduction

Parvovirus B19 infection (B19V) is often asymptomatic or presents with mild disease. It can present as erythema infectiosum during childhood, a viral exanthem (also known as “slapped cheek” syndrome), hydrops fetalis in pregnant women, which can lead to miscarriages, or with less common symptoms that include painful or swollen joints (polyarthopathy syndrome). The infection is usually self-limited and resolves within one to two weeks. Due to red blood cell (RBC) precursors tropism, B19V might cause the temporary cessation of the bone marrow’s RBC production, leading to transient aplastic crisis (TAC), mainly in patients with chronic hemolysis, including hereditary spherocytosis (HS). The overlapping arrest of RBC production and excessive destruction can cause potentially life-threatening anemia, requiring urgent blood transfusions [1].

Pure red cell aplasia (PRCA) has been described in patients living with HIV/AIDS (PLWHA), affecting up to 4.5% of patients [2,3]. Therefore, some authors recommend the prompt consideration of B19V-mediated PRCA for PLWHA with severe isolated anemia [4]. Patients with persistent or recurrent viremia have absent or low levels of specific antibodies. Clinical hallmarks of B19V infection in PLWHA include fatigue and pallor, while immune-mediated symptoms (rash and arthralgia) are generally lacking. The treatment of persistent B19V with intravenous immunoglobulin (IVIG) reduces the viral load and usually results in a marked resolution of anemia [5].

Herein, we report three rare cases of Brazilian PLWHA with B19V-mediated severe anemia. The HIV patients were treated at the Instituto Nacional de Infectologia Evandro Chagas (INI/Fiocruz) and manifested with different clinical presentations. We discuss B19V infection in PLWHA, clinical presentation, diagnostic challenges, and therapeutic responses. These cases highlight the importance of the early recognition of B19V disease in unexplained cytopenias settings.

## 2. Detailed Case Descriptions

### 2.1. Case 1

A 27-year-old Black male diagnosed with HIV months after birth in 1994 had a history of multiple infections, including pneumocystosis, tuberculosis (TB), and bacterial pneumonia. Due to poor adherence to antiretroviral therapy (ART), the patient had an extensively drug-resistant viral infection and was taking lamivudine (3TC), tenofovir (TDF), darunavir/ritonavir (DRV/r), dolutegravir (DTG), etravirine (ETR), and fostemsavir, and presented with anemia and mild leukopenia in November 2020. A bone marrow biopsy showed 70% cellularity, normoblastic erythroid series, reduced granulocytic series, and megakaryocytes with preserved morphology, absence of granulomas, and negative special stains for fungi and mycobacteria. Malignancies were ruled out. In April 2021, he complained of vertigo, headache, vomiting, and fever, and was admitted with a deterioration of the anemic condition, mild hepatitis, and no adenopathy. He exhibited hemoglobin (Hb) 1.7 g/dL, hematocrit (Ht) 6.5%, white blood cells (WBC) 2.33 × 10^9^/L, neutrophils 0.204 × 10^9^/L, and platelets 354,000/mm^3^, with corrected reticulocytes 0%, aspartate aminotransferase 251 U/L, alanine aminotransferase 351 U/L, HIV viral load 40,305 copies/mL, and a CD4^+^ count of 6 cells/µL (Table 1). During hospitalization, he received a transfusion of six packed RBCs. He returned in October 2021 with complaints of dyspnea on minor exertion and intense prostration and reported chest pain the night before. He presented tachycardia and hepatosplenomegaly and received multiple blood transfusions. The myelogram was reviewed, showing giant proerythroblasts with nuclear inclusions (Figure 1A), which raised suspicions of B19V infection. B19V screening included nested PCR for the VP1/VP2 genome region [6], with viral load determined by real-time polymerase chain reaction (qPCR) for the NS1 genome region [7]. In October 2021, B19V PCR was positive in serum with 4.3 × 10^10^ IU/mL, while anti-B19V IgM and IgG were negative. Genotype 1a was defined based on a B19V phylogenetic tree (Figure 2). He received IVIG 400 mg/kg/day for three consecutive days and showed a complete response with stabilized RBC counts. In July 2022, the patient remained non-adherent to ART but without anemia, maintaining a low CD4^+^ count (11 cells/µL) and high HIV viral load (27,642 copies/mL). B19V DNA remained detectable (1.8 × 10^6^ IU/mL), and IgM and IgG were indeterminate. As he was asymptomatic with normal blood cell counts, a new course of IVIG was not indicated.

A phylogenetic tree of Parvovirus B19 (VP1/VP2 gene; among 420 bp) in patients with HIV and anemia (red diamonds), defined as genotype 1a, inferred by using the Maximum Likelihood method and Tamura-Nei model [8]. Evolutionary analyses were conducted in MEGA X [9]. A human bocavirus (HBoV; GQ243610) sequence was used to root the tree. Sequences were analyzed using the Bio-Edit sequence alignment editor, v. 7.2.5 (mbio.ncsu.edu/BioEdit/bioedit.html, accessed on 17 July 2022) and they were compared with the following sequences available in GenBank (Hall 1999): Genotype 1a: EF089179 and EF089209 (Brazil/PA), KC013321 and KC013325 (Brazil/SP), Z68146-Stu and EU478578-EU478562 (Germany), M13178-Au (USA), AF162273-Hv (Finland), M24682-Wi (UK), U38508 (Ireland), U38546-BrIII (Brazil/RJ), DQ293995 (Belgium), JN211168 (The Netherlands); Genotype 1b: DQ357064 and DQ357065 (Vietnam); Genotype 2: DQ333426, AY903437 and AJ717293-Berlin (Germany), AY064475-A6 and AY064776 (Italy), AY044266-LaLi (Finland); Genotype 3a: AJ249437-V9 (France), AY582125, DQ234775, DQ234769 and DQ234771 (Ghana); and Genotype 3b: AY083234-D91.1 (France), DQ408302-DQ408305 (Germany), AY582124, DQ234778-DQ234779 (Ghana). The three samples from this study were deposited in the GenBank database under accession numbers OQ917657, OQ917658, and OQ917659.

### 2.2. Case 2

A 61-year-old White male of Spanish origin living with HIV since 2009 with regular use of TDF, 3TC, and DTG since 2017 presented an undetectable HIV viral load and historically low CD4^+^ counts below 200 cells/mm^3^. He sought medical consultation in August 2021 after four weeks of malaise, lethargy, and an acute incident of dizziness, diarrhea, and syncope. He had hypothyroidism and no history of hematologic disorders, but his mother suffered from chronic anemia. The patient was submitted to a full blood workup that showed pancytopenia (Hb 5.6 g/dL, Ht 15.2%, WBC 1.48 × 10^9^/L, neutrophils 0.547 × 10^9^/L, and platelets 41,000 per mm^3^), mildly elevated serum ferritin (611 μg/L), and anemic hemolytic features (Table 1). He also presented a negative direct antiglobulin test and did not show evidence of renal dysfunction, iron, or glucose-6-phosphate dehydrogenase deficiency. Reticulocyte counts were not available. During the physical examination, he exhibited jaundice and a palpable spleen. He was admitted for further investigation and received three transfusions of packed RBCs. The HIV viral load at admission was undetectable, and the CD4^+^ count was 127 cells/mm^3^. As he sustained pancytopenia, a bone marrow aspirate and a biopsy were performed. The myelogram showed giant proerythroblasts with nuclear inclusions (Figure 1B). The bone marrow histology disclosed 95% cellularity, mild erythroid hyperplasia, megaloblastic findings, and no hemophagocytic lymphohistiocytosis. Furthermore, the bone marrow sample showed negative results in the acid-alcohol-fast-bacilli (AFB) test, cultures for fungi, and mycobacteria. Analyses of anti-B19V IgM were indeterminate, and anti-IgG antibodies were positive. Results from the PCR included the detection of 8.92 × 10^3^ IU/mL B19V, and genotype 1a was identified (Figure 2). The patient received IVIG 500 mg/kg/d for four days after 37 days from the onset of symptoms. The patient had no significant symptoms, and the RBC count stabilized immediately after treatment. Ten months later, his laboratory results showed a resolution of the pancytopenia but the maintenance of a mild elevation of indirect bilirubin, indicative of chronic hemolysis. B19V PCR was no longer detected in peripheral blood. Abdominal ultrasonography excluded gallstones or the maintenance of splenomegaly. A blood smear analysis was performed, which revealed spherocytes in more than 50% of RBCs (Figure 1C), and an osmotic fragility test (OFT) was suggestive of RBC membranopathy. In addition, hemoglobin electrophoresis showed no changes. We concluded that this case was B19V-induced pancytopenia in an immunocompromised patient presenting with HS. He continued receiving folate supplementation and had no clinical symptoms.

### 2.3. Case 3

A 27-year-old Black man was newly diagnosed with HIV in December 2021 during admission due to respiratory symptoms and pancytopenia. He also presented disseminated TB and had increased transaminase levels after starting treatment with rifampicin, isoniazid, pyrazinamide, and ethambutol. Thus, doctors suspected TB-related hepatotoxicity and switched treatment to levofloxacin, amikacin, and ethambutol, while he tolerated the reintroduction of the first-line scheme over three weeks. The patient presented with 345,605 copies/mL HIV viral load and 69 cells/mm^3^ CD4^+^ counts. After four weeks of TB treatment, ART was initiated with a negative AFB in the sputum. Eighteen days after starting ART, he presented with a new asymptomatic elevation of transaminases associated with a pattern of a cholestatic lesion, anemia (Hb 9.4 d/dL and Ht 29.1%), and leukopenia (WBC 2510 cells/mm^3^) and average platelet counts. One month after ART initiation—on February 2022—the patient was hospitalized with anemia aggravation requiring RBC transfusion and cholestatic hepatitis, leading to a new suspension of tuberculostatic drugs. The HIV viral load decreased to 130 copies/mL, and CD4^+^ counts improved to 138 cells/mm^3^ (Table 1). The patient was re-screened for opportunistic infections, with a negative investigation for histoplasmosis and cryptococcosis. A myelogram showed enlarged erythroblasts with nuclear inclusions (Figure 1E). Bone marrow histopathologic analysis showed serous stromal degeneration and a reactive lymphoid aggregate. Abdominal ultrasonography evidenced hepatosplenomegaly and the thickening of the gallbladder wall with no signs of lithiasis. To address the TB-associated inflammatory reconstitution inflammatory syndrome (IRIS-TB) hypothesis, he received systemic corticosteroid therapy (prednisone 1 mg/kg/day for seven days with posterior waning) and presented a resolution of laboratory abnormalities. He tolerated the reintroduction of anti-TB treatment. A retrospective analysis from a serum sample collected in February 2022 revealed B19V DNA detectable at 5.9 × 10^4^ IU/mL, B19V genotype 1a, and anti-B19V IgG positivity, corroborating bone marrow findings and confirming the B19V infection (Table 1 and Figure 2). No additional measures were adopted for B19V, as the patient’s DNA became undetectable, and symptoms resolved after IRIS-TB treatment.

## 3. Discussion

The relationship between anemia and B19V infection is well known [4,10,11,12]. B19V DNA was detected by dot blot hybridization in sera from 5 of 30 (17%) PLWHA with Ht <24% and 4 of 13 (31%) patients with Ht < 20% in a study published in 1997, suggesting that B19V was a substantial contributor to severe anemia in HIV infection in the pre-high-potency ART era [13]. A recent study that enrolled 158 HIV-infected children in Nigeria showed a low prevalence of B19V among HIV-positive children (~2%). Nevertheless, a significant relationship was established between B19V infection and the severity of anemia (*p* = 0.015) [14]. A Brazilian study conducted ten years ago estimated a frequency of B19V seroconversion of 31.8% in a cohort of 88 HIV-infected patients, and showed that patients who seroconverted were 5.40 times more likely to have anemia than those who did not [15]. There is a lack of more recent data on the frequency of B19V and HIV coinfection in Brazil.

Adherence to medication is one of the most critical factors for a successful ART. Poor medication adherence can lead to treatment failure and the development of drug-resistant strains of HIV. Several studies have shown that adherence to antiretroviral medication is associated with better health outcomes in HIV-positive individuals [16,17]. Hematological changes are common findings in PLWHA, particularly in poor adherent individuals. Anemia and thrombocytopenia have been demonstrated to be independent predictors of morbidity and mortality [18]. Although cytopenias often respond to ART, in some patients they can persist. Our case series confirmed that the B19V course depends on the host’s immunologic status, as DNA detection only lasted in the first case, who was a patient not adherent to ART.

Several case reports describe B19V infection in patients with HS, with patient’s findings usually including fever, fatigue, a family history of HS in most patients (59%), and in some cases liver dysfunction [19]. Although HS diagnosis usually occurs in childhood and young adult life, HS may be diagnosed at any time, including in old age [20]. B19V infection can cause TAC in patients with increased destruction or loss of RBCs, as it occurs in HS. Patients may also have congestive heart failure, hypo flow stroke, and acute splenic sequestration [21]. Our second case’s clinical presentation and evolutive findings after IVIG suggested TAC with an underlying chronic hemolytic anemia. HS was confirmed after investigation.

HS is the most common RBC membrane disorder worldwide, and the most common hereditary hemolytic anemia in people with Northern European ancestry, with a prevalence of 1 in 1000 to 2500 [22]. The abnormal spherocytes’ shape is due to inadequate vertical linkages between the cytoskeleton and the lipid bilayer [22]. RBCs exhibit increased osmotic fragility and deformability, resulting in extravascular hemolysis due to the increased destruction of RBCs when passing through the spleen. The eosin-5′-maleimide binding test is the most accurate screen for HS diagnosis because it binds to specific erythrocyte membrane molecules [23]. However, it is scarcely offered in Brazil and is only required in complex cases. The diagnosis here was straightforward based on clinical history, physical examination, and laboratory data. Although TAC specifically affects RBC lines, WBCs and platelets may also decline. Despite high suspicion, we could not characterize TAC in our second patient due to the lack of reticulocyte counts and typical histologic bone marrow findings during the pancytopenia episode.

Bone marrow potentially suffers from the combined effects of HIV, inflammatory mediators released during infection, nutritional deficiency, and opportunistic pathogens [24]. B19V should be considered part of the differential diagnosis of severe hypoproliferative anemia in PLWHA. B19V diagnosis is not obvious, and clinicians should search for bone marrow abnormalities and molecular virological findings. Due to immunosuppression, PLWHA usually lack IgM production, leading to the prolonged destruction of RBC progenitors [25]. B19V might have played a role in our third case’s presentation, despite the patient having other possible causes for anemia and hepatitis, such as HIV myelopathy, opportunistic infections, TB-related hepatotoxicity, and IRIS-TB.

Nevertheless, the diagnosis of B19V-induced anemia in PLWHA is rare, possibly due to underdiagnosing [25,26]. Molecular diagnosis through qPCR was decisive in establishing B19V diagnosis in our cases because of the low serologic sensitivity. Genotype 1a was found in all samples of this study, and there was no apparent relationship between the infecting genotype and the clinical course. Genotype 1 is the most common B19V genotype detected globally [27,28] and among HIV patients [14]. In a Brazilian study among five HIV-positive patients receiving ART who had B19V infections confirmed by qPCR, four exhibited genotype 1a strain, and the remaining patient presented a genotype 3b strain [29]. Ferry et al. found that the detection of B19V genotypes 2 or 3 was infrequent in a large cohort of immunocompromised, HIV-infected anemic patients, despite the use of highly sensitive qPCR methods [30]. Although these three genotypes have been reported in Brazil, genotype 1 is the most common [31,32,33]. The B19V sequences obtained from three cases were closely related to sequences EF089179, EF089209, and KC013325, isolated from Brazilian patients (from Rio de Janeiro, São Paulo and Pará states, respectively) with compromised immunological and/or hematological status [31,34].

Despite the low CD4^+^ counts, our patients did not experience a worse prognosis than immunocompetent individuals. The treatment of B19V infection depends on the clinical burden and the host’s immune response. Spontaneous resolution can occur if the immune system reconstitutes; no further treatment is needed in these cases. Still, IVIG is a fundamental source of neutralizing antibodies for persistently symptomatic individuals. Usually, only one course of IVIG is required for long-term remission [5]. The coronavirus disease 2019 (COVID-19) resulted in interruptions in HIV diagnosis and virological control, impacting ART supply chains and negatively affecting PLWHA’s treatment adherence and quality of care [35]. We hypothesize that the COVID-19 pandemic may have influenced B19V’s incidence in PLWHA. Further studies regarding B19V frequency in PLWHA are needed in Brazil to understand factors associated with severity and disease evolution, particularly after the COVID-19 pandemic.

## 4. Conclusions

Our three cases descriptions highlight the importance of B19V diagnosis in PLWHA. Cytopenias are often multifactorial, and performing a comprehensive clinical evaluation is essential to prevent delayed diagnosis and morbidity. Persistent B19V infection in PLWHA is challenging to differentiate from other opportunistic infections or IRIS. Therefore, a high index of suspicion for B19V should be of concern for HIV patients with cytopenias and advanced immunosuppression. The decision to treat B19V should always be clinically guided. Although IVIG therapy is the primary treatment to clear viremia in PLWHA, ART compliance is essential to normalize blood cell counts and prevent B19V relapse.

## Figures and Tables

**Figure 1 viruses-15-01124-f001:**
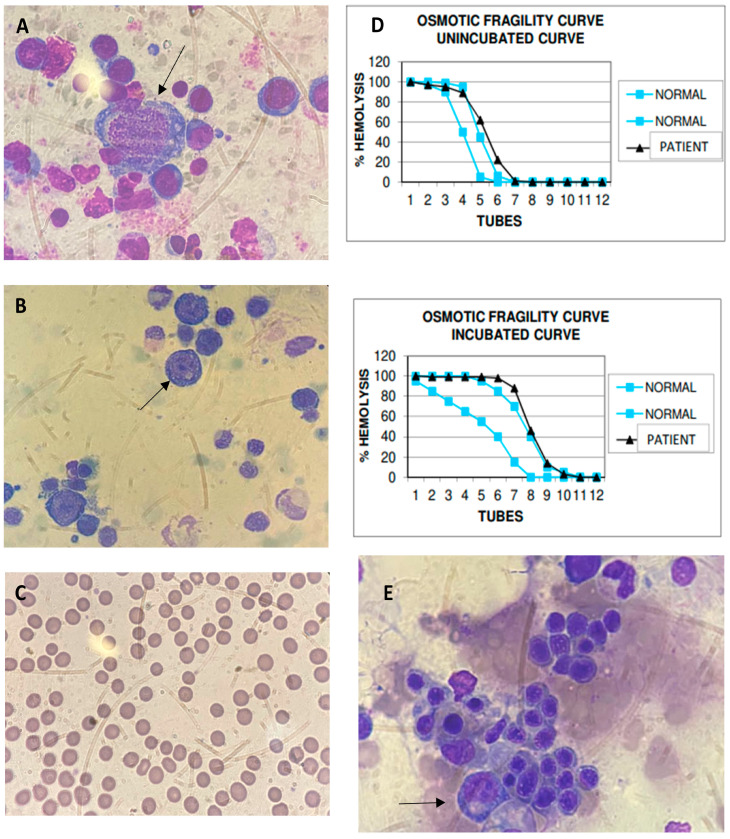
Myelogram, peripheral blood smear, and osmotic fragility test curve findings in patients with B19V and HIV. B19V inclusion in abnormally large pronormoblast with basophilic and vacuolated cytoplasm in bone marrow aspirate (arrows) (×400) is shown for Cases 1 (**A**) and 2 (**B**). Small-sized, spherical-shaped, deep-staining red blood cells (RBC) are shown to lack an area of central pallor at peripheral blood film, accounting for more than 50% of RBC in Case 2 (**C**). (**D**) The osmotic fragility test curve (black) for Case 2 is shown; slightly right-skewed curves suggest that RBCs hemolyze more quickly than usual (blue curves as range reference) both in unincubated and in incubated curves. B19V inclusion in abnormally large pronormoblast with basophilic and vacuolated cytoplasm in bone marrow aspirate is shown for Case 3 (arrow) (×400) (**E**).

**Figure 2 viruses-15-01124-f002:**
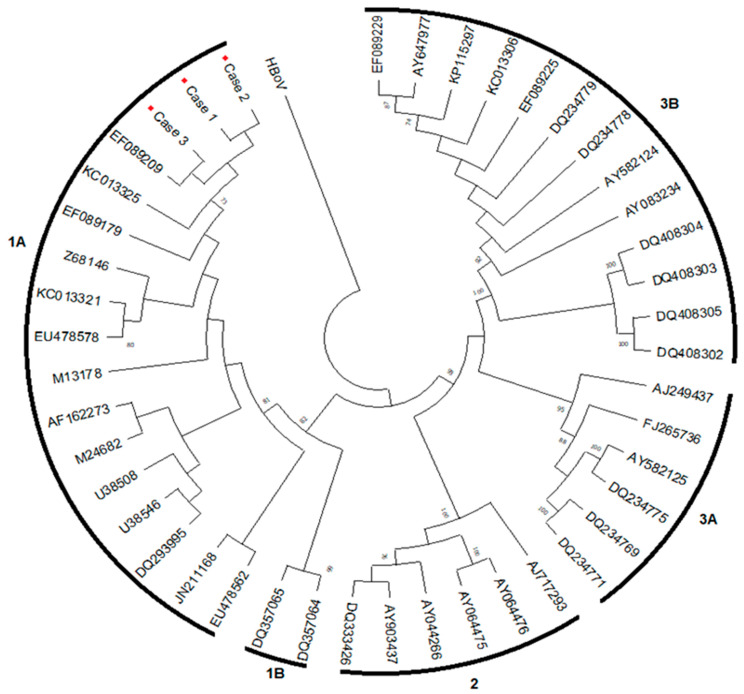
Parvovirus B19 phylogenetic tree.

**Table 1 viruses-15-01124-t001:** Laboratory findings’ evolution of three HIV-positive patients with Parvovirus B19 infection.

			Case 1			Case 2			Case 3		
Laboratory Analysis	Test Method	Reference Value	April/21 *	October/21	July/22	April/21	August/21 *	June/22	January/22	February/22 *	July/22
Hemoglobin (g/dL)	Flow Cytometry	(10.5–14.8)	1.7	5.2	13.9	12.5	5.6	16.4	9.4	6.5	11.6
Hematocrit (%)	Flow Cytometry	(40–54)	6.5	15.6	40.3	34.8	15.2	46.5	29.1	18.7	35.3
MCV (µm^3^)	Flow Cytometry	(80–96)	85.0	82.0	85.4	111.5	104.8	101.3	82.4	77.3	82.7
MCHC (g/dL)	Flow Cytometry	(32–36)	26.0	33.0	34.5	35.9	36.8	35.3	26.6	26.9	27.2
Platelet counts (per mm^3^)	Flow Cytometry	(155,000–409,000)	354,000	255,000	143,000	119,000	41,000	135,000	245,000	175,000	241,000
WBC (per mm^3^)	Flow Cytometry	(4500–11,000)	2330	2790	3950	4780	1480	5890	2510	1190	2880
Neutrophil counts (per mm^3^)	Flow Cytometry	(1470–6750)	204	1850	2528	3435	577	3239	1832.3	654.4	2163
Direct antiglobulin	Gel Test	Negative	Negative	Negative	Positive	NA	Negative	Negative	NA	NA	Negative
G6PD phenotype	Brewer	Normal	NA	NA	Normal	NA	Normal	Normal	NA	NA	NA
Reticulocytes **	Flow Cytometry	(0.5–1.5%)	0.0	0.0	1.1	NA	NA	0.2	0.8	ND	ND
Lactate desidrogenase (UI/L)	Enzymatic	(85–227)	243	448	218	209	6204	179	962	830	518
Ferritin (ng/mL)	Immunoassay	(26–388)	1215	1302	701	NA	611	596	3078	4037	ND
C-reactive protein (mg/dL)	Turbidimetric	(<0.3)	5.2	23.73	2.52	0.63	1.27	NA	1.0	3.45	0.6
Aspartat aminotransferase (U/L)	Enzymatic	(15–37)	251	26	22	114	190	26	93	455	74
Alanine aminotransferase (U/L)	Enzymatic	(12–78)	302	51	31	62	38	25	159	414	99
Total bilirrubin (mg/dL)	Jendrassik and Grof	(0.0–1.0)	0.85	1.36	0.6	0.93	3.07	1.38	0.65	1.21	0.80
Indirect bilirrubin (mg/dL)	Jendrassik and Grof	(0.0–0.7)	0.47	0.91	0.14	0.69	2.27	0.97	0.33	0.38	0.36
GGT (U/L)	Enzymatic	(15–85)	122	118	37	80	40	152	368	626	628
Alkaline phosphatase (U/L)	Enzymatic	(46–116)	1140	133	104	69	61	64	665	1529	326
HIV viral load (copies/mL)	qPCR	ND	40,305	NA	27,642	ND	ND	ND	345,605	130	72
CD4+ T lymphocytes (cells/µL)	Flow Cytometry	(40.4–1612)	6	65	11	NA	127	200	69	138	299
PCR VP1VP2	PCR	-	Positive	NA	Positive	NA	Positive	Negative	NA	Positive	Negative
qPCR NS1 (IU/mL)	qPCR	-	4.3 × 10^10^	NA	1.8 × 10^6^	NA	8.9 × 10^3^	Negative	NA	5.9 × 10^4^	Negative
IgM anti-B19V	ELISA	-	Negative	NA	ID	NA	ID	ID	NA	Negative	Negative
IgG anti-B19V	ELISA	-	Negative	NA	ID	NA	Positive	Positive	NA	Positive	Positive

Abbreviations: WBC: White blood cell; MCV: Mean Corpuscular Volume; MCHC: Mean corpuscular hemoglobin concentration; B19V: Parvovirus B19; G6PD: Glucose-6-phosphate dehydrogenase; GGT: Gamma-glutamyl transpeptidase; ND: Not detectable; NA: Not available; ID: indeterminate; PCR: Polymerase chain reaction; qPCR: Real time polymerase chain reaction; ELISA: Enzyme Linked Immuno-Sorbent Assay; * B19V detection; ** Reticulocytes corrected by the level of anemia.

## Data Availability

The data that support the findings of this study are available from the National Center for Biotechnology Information (NCBI) at https://www.ncbi.nlm.nih.gov/.

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
