# Peer review of "Clinical Presentation of Parvovirus B19 Infection in Adults Living with HIV/AIDS: A Case Series"

_viruses, 2023, doi:10.3390/v15051124_

Round 1

Reviewer 1 Report

The manuscript titled “Clinical Presentation of Parvovirus B19 Infection in Adults Living with HIV/AIDS: A Case Series” aimed at establishing the importance of detecting B19V infection early in in HIV patients presenting with cytopenia and advanced immunosuppression. Three unique cases were thoroughly described, and the evidence provided to support their aim was convincing. The discussion as outlined from the authors is well presented and indeed does support the importance of testing through PCR for B19V in these patients.

Minor considerations:

*Figure 1 D) caption does not detail from which case 1, 2 or 3 the data comes from. Please mention also in the legend this

*Figure 1 C is difficult to see and magnification is not provided

*References are not according to author’s instructions. Please update the manuscript accordingly to the proper way to cite in this journal.

*The following sentence uses incorrect punctuation and should be updated accordingly. “A bone marrow biopsy showed 70% cellularity; normoblastic erythroid series; reduced granulocytic series; megakaryocytes with preserved morphology and absence of granulomas, with negative special stains for fungi and mycobacteria, and ruled out malignancies.

Author Response

We are re-submitting a revised manuscript entitled “Clinical Presentation of Parvovirus B19 Infection in Adults Living with HIV/AIDS: A Case Series” by Daniela P. Mendes-de-Almeida, Joanna Bokel, Arthur Daniel Rocha Alves, Alexandre G. Vizzoni, Isabel Cristina Ferreira Tavares, Mayara Secco Torres Silva, Juliana dos Santos Barbosa Netto, Beatriz Grinsztejn, and Luciane Almeida Amado Leon*.  We appreciate the time and effort that you and the reviewers dedicated to providing feedback, based on which we have revised the manuscript and feel this has resulted in an improved version. The modifications are highlighted (in yellow color) in the manuscript, and point-by-point responses to reviewers’ comments are provided below.

Reviewer comments:

Reviewer #1:

Comments and Suggestions for Authors

The manuscript titled “Clinical Presentation of Parvovirus B19 Infection in Adults Living with HIV/AIDS: A Case Series” aimed at establishing the importance of detecting B19V infection early in in HIV patients presenting with cytopenia and advanced immunosuppression. Three unique cases were thoroughly described, and the evidence provided to support their aim was convincing. The discussion as outlined from the authors is well presented and indeed does support the importance of testing through PCR for B19V in these patients.

Minor considerations:

 *Figure 1 D) caption does not detail from which case 1, 2 or 3 the data comes from. Please mention also in the legend this.

Reply: Thanks for point this out. The information was added in the legend.

*Figure 1 C is difficult to see and magnification is not provided

 Reply: We improved this figure’s quality to 600 dpi.

*References are not according to author’s instructions. Please update the manuscript accordingly to the proper way to cite in this journal.

Reply:  The references were modified according to the journal website.

*The following sentence uses incorrect punctuation and should be updated accordingly. “A bone marrow biopsy showed 70% cellularity; normoblastic erythroid series; reduced granulocytic series; megakaryocytes with preserved morphology and absence of granulomas, with negative special stains for fungi and mycobacteria, and ruled out malignancies.”

Reply: As requested by the reviewer, the sentence was changed to “A bone marrow biopsy showed: 70% cellularity; normoblastic erythroid series; reduced granulocytic series; and megakaryocytes with preserved morphology, absence of granulomas and negative special stains for fungi and mycobacteria. Malignancies were ruled out.”, lines 70-73.

Reviewer 2 Report

The manuscript entitled “Clinical Presentation of Parvovirus B19 Infection in Adults Liv-2 ing with HIV/AIDS: A Case Series” by Almedia and co-authors reported three cases of B19V infection diagnosed by real-time polymerase chain reaction in adult’s human with HIV. The first patients had sever anemia and low CD4 counts. While, second patient had low CD4 count. Third patient diagnosed with HIV and TB diagnosed TB-treatment hepatoxicity. Overall, the case description enlightens the role of B19V diagnosis in HIV/AIDS. The manuscript is written well and all the figures are relevant to the work. Although, I have a few queries that need to be addressed prior to consideration of manuscript for publication. I recommend a minor revision of the manuscript. Here are my queries/suggestions for the manuscript:-

1.      Authors need to reframe the abstract.

2.      Manuscript need to screen for typos and grammatical errors.

3.      Authors need to discuss the outcomes of latest studies.

4.      Authors should share the high resolution images in revised version of manuscript.

The manuscript entitled “Clinical Presentation of Parvovirus B19 Infection in Adults Liv-2 ing with HIV/AIDS: A Case Series” by Almedia and co-authors reported three cases of B19V infection diagnosed by real-time polymerase chain reaction in adult’s human with HIV. The first patients had sever anemia and low CD4 counts. While, second patient had low CD4 count. Third patient diagnosed with HIV and TB diagnosed TB-treatment hepatoxicity. Overall, the case description enlightens the role of B19V diagnosis in HIV/AIDS. The manuscript is written well and all the figures are relevant to the work. Although, I have a few queries that need to be addressed prior to consideration of manuscript for publication. I recommend a minor revision of the manuscript. Here are my queries/suggestions for the manuscript:-

1.      Authors need to reframe the abstract.

2.      Manuscript need to screen for typos and grammatical errors.

3.      Authors need to discuss the outcomes of latest studies.

4.      Authors should share the high resolution images in revised version of manuscript.

Author Response

We are re-submitting a revised manuscript entitled “Clinical Presentation of Parvovirus B19 Infection in Adults Living with HIV/AIDS: A Case Series” by Daniela P. Mendes-de-Almeida, Joanna Bokel, Arthur Daniel Rocha Alves, Alexandre G. Vizzoni, Isabel Cristina Ferreira Tavares, Mayara Secco Torres Silva, Juliana dos Santos Barbosa Netto, Beatriz Grinsztejn, and Luciane Almeida Amado Leon*.  We appreciate the time and effort that you and the reviewers dedicated to providing feedback, based on which we have revised the manuscript and feel this has resulted in an improved version. The modifications are highlighted (in yellow color) in the manuscript, and point-by-point responses to reviewers’ comments are provided below.

Reviewer comments:

Comments and Suggestions for Authors

The manuscript entitled “Clinical Presentation of Parvovirus B19 Infection in Adults Living with HIV/AIDS: A Case Series” by Almeida and co-authors reported three cases of B19V infection diagnosed by real-time polymerase chain reaction in adult’s human with HIV. The first patient had severe anemia and low CD4 counts. While the second patient had low CD4 count. Third patient diagnosed with HIV and TB diagnosed TB-treatment hepatoxicity. Overall, the case description enlightens the role of B19V diagnosis in HIV/AIDS. The manuscript is written well and all the figures are relevant to the work. Although, I have a few queries that need to be addressed prior to consideration of manuscript for publication. I recommend a minor revision of the manuscript. Here are my queries/suggestions for the manuscript:-

  1. Authors need to reframe the abstract.

Reply: As suggested, the abstract was modified and improved accordingly:

“Abstract: Parvovirus B19 (B19V) infection varies clinically depending on the host's immune status. Due to red blood cell precursors tropism, B19V can cause chronic anemia and transient aplastic crisis in patients with immunosuppression or chronic hemolysis. We report three rare cases of Brazilian adults living with human immunodeficiency virus (HIV) with B19V infection. All cases presented severe anemia and required red blood cell transfusions. The first patient had low CD4+ counts and was treated with intravenous immunoglobulin (IVIG). As he remained poor adherent to antiretroviral therapy (ART), B19V detection persisted. The second patient had sudden pancytopenia despite being on ART with an undetectable HIV viral load. He had historically low CD4+ counts, fully responded to IVIG, and had undiagnosed hereditary spherocytosis. The third individual was recently diagnosed with HIV and tuberculosis (TB). One month after ART initiation, he was hospitalized with anemia aggravation and cholestatic hepatitis. Analysis of his serum revealed B19V DNA and anti-B19V IgG, corroborating bone marrow findings and a persistent B19V infection. The symptoms resolved and B19V became undetectable. In all cases, real time PCR was essential for diagnosing B19V. Our findings showed that adherence to ART was crucial to B19V clearance in HIV-patients and highlighted the importance of early recognition of B19V disease in unexplained cytopenias.”

  1. Manuscript need to screen for typos and grammatical errors.

Reply: The manuscript was revised by a native English-speaking professional.

  1. Authors need to discuss the outcomes of latest studies.

Reply: We added the discussion of the importance of ART adherence to HIV virological control: “Adherence to medication is one of the most critical factors for the success of ART treatment. Poor medication adherence can lead to treatment failure and the development of drug-resistant strains of HIV. Several studies have shown that adherence to antiretroviral medication is associated with better health outcomes in HIV-positive individuals. [16,17]. Hematological changes, such as anemia or thrombocytopenia, are common findings in PLWHA, particularly in poor adherent individuals. Anemia and thrombocytopenia have been demonstrated to be independent predictors of morbidity and mortality [18]”, lines 200-206, page 4 and included the sentence “We hypothesize that the COVID-19 pandemic may have influenced B19V's incidence in PLWHA”, lines 266 and 267, page 4.

  1. Authors should share the high-resolution images in revised version of manuscript.

Reply: We improved images quality to 600 dpi.

Reviewer 3 Report

The article summarise cases of Parvovirus B19 infection in three HIV/AIDS patient. The cases are well described, the development of the disease can be followed with the very informative Table 1 and the text.

Although it is a Case Report, some methodological details should be clarified and expanded more detailed. The easiest way would be inserting an extra column in Table 1 mentioning the test method/kit or the device or reference of the method. Or you should edit these pieces of information into the text.

A Figure should be understandable with its legend alone, please clarify which gene, how long segment was analysed. Moreover, there is not dedicated methodological paragraph for sequencing and phylogenetic analyses, therefore several other details should be listed in legend of Figure 2. How and from where have you selected the other sequences included in the analysis? What is the reference of the clustering? 

In Discussion it would be interesting to mention anything about the origin of the sequences closely related to the new ones (EF089209; KC013325, EF089179). They are also Brazilian sequences, but this fact is not even mentioned in the article. You should discuss their relevance and connection to the reported cases. What are the similarity values among the three new sequences as well as among them and the other Brazilians? Do they form a distinct group or  might other sequences be inserted among them?

In line 230 you mention: 'this is the predominant strain'. What strain do you refer to? Please involve it into the tree and the Discussion. You should dedicate a paragraph in Discussion on this topic. Is there any relevant similarities, connection to the reported cases? Any specialities of B19 strains revealed from HIV patients?

The three parvovirus sequences should be uploaded in GenBank and you should refer to their Acc. No. in the article.

In Fig. 1D the osmotic fragility curves of the patient are closer to one of the 'normal' ones than the difference between the two normal curves. How can you explain it? Why does it demonstrates any pathological deviation?

Some minor mistakes, questions:

In Case 1 CD4+ count is mentioned as 6 cells/ul in the text, but 2 in the Table. Which one is the correct value?

Dimensions of HIV viral load and CD4+ are missing from Table 1.

Use copies/mL instead of cps/mL!

The English Language usage is rather good, the text is informative. However, a native speaker may need to check it.

Author Response

We are re-submitting a revised manuscript entitled “Clinical Presentation of Parvovirus B19 Infection in Adults Living with HIV/AIDS: A Case Series” by Daniela P. Mendes-de-Almeida, Joanna Bokel, Arthur Daniel Rocha Alves, Alexandre G. Vizzoni, Isabel Cristina Ferreira Tavares, Mayara Secco Torres Silva, Juliana dos Santos Barbosa Netto, Beatriz Grinsztejn, and Luciane Almeida Amado Leon*.  We appreciate the time and effort that you and the reviewers dedicated to providing feedback, based on which we have revised the manuscript and feel this has resulted in an improved version. The modifications are highlighted (in yellow color) in the manuscript, and point-by-point responses to reviewers’ comments are provided below.

Reviewer comments:

The article summarize cases of Parvovirus B19 infection in three HIV/AIDS patient. The cases are well described, the development of the disease can be followed with the very informative Table 1 and the text.

Although it is a Case Report, some methodological details should be clarified and expanded more detailed. The easiest way would be inserting an extra column in Table 1 mentioning the test method/kit or the device or reference of the method. Or you should edit these pieces of information into the text.

Reply: As suggested, we added a column containing test methods in Table 1.

A Figure should be understandable with its legend alone, please clarify which gene, how long segment was analysed. Moreover, there is not dedicated methodological paragraph for sequencing and phylogenetic analyses, therefore several other details should be listed in legend of Figure 2. How and from where have you selected the other sequences included in the analysis? What is the reference of the clustering? 

Reply: Thank you for pointing this out. We analysed VP1/VP2 partial gene with 420bp. The legend of the figure was improved with the following statement (lines 108-122)

Phylogenetic tree of Parvovirus B19 (VP1/VP2 gene; among 420bp) in patients with HIV and anemia (red diamonds), defined as genotype 1a, inferred by using the Maximum Likelihood method and Tamura-Nei model (Tamura K & Nei M, 1993). Evolutionary analyses were conducted in MEGA X (Kumar et al, 2018). A Human bocavirus (HBoV; GQ243610) sequence was used to root the tree. Sequences were analyzed by the Bio-Edit sequence alignment editor, v. 7.2.5 (mbio.ncsu.edu/BioEdit/bioedit.html) and they were compared with the following sequences available in GenBank (Hall 1999): Genotype 1a: EF089179 and EF089209 (Brazil/PA), KC013321 and KC013325 (Brazil/SP), Z68146-Stu and EU478578-EU478562 (Germany), M13178-Au (USA), AF162273-Hv (Finland), M24682-Wi (UK), U38508 (Ireland), U38546-BrIII (Brazil/RJ), DQ293995 (Belgium), JN211168 (Netherland); Genotype 1b: DQ357064 and DQ357065 (Vietnam); Genotype 2: DQ333426, AY903437 and AJ717293-Berlin (Germany), AY064475-A6 and AY064776 (Italy), AY044266-LaLi (Finland); Genotype 3a: AJ249437-V9 (France), AY582125, DQ234775, DQ234769 and DQ234771 (Ghana); Genotype 3b: AY083234-D91.1 (France), DQ408302-DQ408305 (Germany), AY582124, DQ234778-DQ234779 (Ghana). The three samples from this study were deposited in GenBank database under accession numbers XX123456, YY123456 and ZZ123456.

In Discussion it would be interesting to mention anything about the origin of the sequences closely related to the new ones (EF089209; KC013325, EF089179). They are also Brazilian sequences, but this fact is not even mentioned in the article. You should discuss their relevance and connection to the reported cases. What are the similarity values among the three new sequences as well as among them and the other Brazilians? Do they form a distinct group, or might other sequences be inserted among them?

Reply: We improved our discussion with the following statement: “The B19 sequences obtained from three cases was closely related to sequences EF089179, EF089209 and KC013325, isolated from Brazilian patients (from Rio de Janeiro, São Paulo and Pará state, respectively) with compromised immunological and/or hematological status [31,34].”, lines 252-255.

In line 230 you mention: 'this is the predominant strain'. What strain do you refer to? Please involve it into the tree and the Discussion. You should dedicate a paragraph in Discussion on this topic. Is there any relevant similarities, connection to the reported

Reply: In line 230 we refer to genotype 1a, that was detected in our samples B19V positive. As suggested by the reviewer, a paragraph was added in the discussion on this topic: “In a Brazilian study among five HIV-positive patients receiving antiretroviral treatment who had B19V infections confirmed by qRT-PCR, four patients exhibited genotype 1a strain, and the remaining patient presented a genotype 3b strain [29]. Ferry et al found that the detection of B19V genotypes 2 or 3 was infrequent in a large cohort of immunocompromised, HIV-infected anemic patients, despite the use of highly sensitive qRT-PCR methods [30]. Although these three genotypes have been reported in Brazil, genotype 1 is the most common [31–33]”, lines 246-253.

Is there any relevant similarities, connection to the reported cases? Any specialities of B19 strains revealed from HIV patients?

Reply: We added in the discussion similarities with other Brazilian B19V cases, comparing our cases with the sequences KC013325, from a leukemia patient from São Paulo and the two EF089179 and EF089209, from Pará with anemia: “The B19 sequences obtained from three cases was closely related to sequences EF089179, EF089209 and KC013325, isolated from Brazilian patients (from Rio de Janeiro, São Paulo and Pará state, respectively) with compromised immunological and/or hematological status [31,34].”, lines 252-255.

The three parvovirus sequences should be uploaded in GenBank and you should refer to their Acc. No. in the article.

Reply: The three samples from this study were deposited in GenBank database under accession numbers XX123456, YY123456 and ZZ123456 (lines 120-122).

In Fig. 1D the osmotic fragility curves of the patient are closer to one of the 'normal' ones than the difference between the two normal curves. How can you explain it? Why does it demonstrate any pathological deviation?

Reply: Low osmotic resistance may lead to intravascular hemolysis, which causes a reduction of RBCs life span. The osmotic fragility curve of RBC not only reflects the average membrane and cytoplasmic properties but may also provide information on the distribution of those properties within the sample. Comparing to normal individuals’ curves (in blue) we can see that the patient´s sample curve (in black) is slightly right skewed - out of the normal range - indicating a tendency to hemolysis (lines 102-104).

 Some minor mistakes, questions:

In Case 1 CD4+ count is mentioned as 6 cells/ul in the text, but 2 in the Table. Which one is the correct value?

Reply: Thanks for pointing this out. The correct CD4 count was 6 cells/ul.

Dimensions of HIV viral load and CD4+ are missing from Table 1. Use copies/mL instead of cps/mL!

Reply: Thanks for pointing this out. All of them were corrected in Table 1.

Comments on the Quality of English Language

The English Language usage is rather good, the text is informative. However, a native speaker may need to check it.

Reply: As indicated above for Reviewer #1, the manuscript was revised by a native English-speaking professional. 
